# Venetoclax Use in Paediatric Haemato-Oncology Centres in Poland: A 2022 Survey

**DOI:** 10.3390/children10040745

**Published:** 2023-04-19

**Authors:** Katarzyna Bobeff, Agata Pastorczak, Zuzanna Urbanska, Walentyna Balwierz, Edyta Juraszewska, Jacek Wachowiak, Katarzyna Derwich, Magdalena Samborska, Krzysztof Kalwak, Iwona Dachowska-Kalwak, Paweł Laguna, Iwona Malinowska, Katarzyna Smalisz, Jolanta Gozdzik, Aleksandra Oszer, Bartosz Urbanski, Maciej Zdunek, Tomasz Szczepanski, Wojciech Mlynarski, Szymon Janczar

**Affiliations:** 1Department of Pediatrics, Oncology and Hematology, Medical University of Lodz, Sporna 36/50, 91-738 Lodz, Poland; 2Department of Pediatric Oncology and Hematology, Institute of Pediatrics, Jagiellonian University Medical College, Wielicka 265, 30-663 Cracow, Poland; 3Department of Pediatric Oncology, Hematology and Transplantology, Poznań University of Medical Sciences, Szpitalna 27/33, 60-572 Poznan, Poland; 4Department of Pediatric Bone Marrow Transplantation, Oncology, and Hematology, Wroclaw Medical University, Borowska 213, 50-556 Wroclaw, Poland; 5Department of Pediatric Oncology and Hematology, Independent Public Children’s Teaching Hospital, Zwirki i Wigury 63A, 02-091 Warsaw, Poland; 6Stem Cell Transplant Center, Department of Clinical Immunology and Transplantology, University Children’s Hospital, Jagiellonian University Collegium Medicum, Wielicka 265, 30-663 Krakow, Poland; 7Department of Pediatric Hematology and Oncology, Medical University of Silesia, 3-go Maja 13-15, 41-800 Zabrze, Poland

**Keywords:** venetoclax, ABT-199, BH3 mimetics, acute lymphoblastic leukaemia, acute myeloid leukaemia, juvenile myelomonocytic leukaemia

## Abstract

Venetoclax, the best established BH3-mimetic, is a practice-changing proapoptotic drug in blood cancers in adults. In paediatrics the data are fewer but exciting results were recently presented in relapsed or refractory leukaemias demonstrating significant clinical activity. Importantly, the in-terventions could be potentially molecularly guided as vulnerabilities to BH3-mimetics were re-ported. Currently venetoclax is not incorporated into paediatric treatment schedules in Poland but it has been already used in patients that failed conventional therapy in Polish paediatric haemato-oncology departments. The aim of the study was to gather clinical data and correlates of all paediatric patients treated so far with venetoclax in Poland. We set out to gather this experience to help choose the right clinical context for the drug and stimulate further research. The questionnaire regarding the use of venetoclax was sent to all 18 Polish paediatric haemato-oncology centres. The data as available in November 2022 were gathered and analysed for the diagnoses, triggers for the intervention, treatment schedules, outcomes and molecular associations. We received response from 11 centres, 5 of which administered venetoclax to their patients. Clinical benefit, in most cases consistent with hematologic complete remission (CR), was reported in 5 patients out of ten, whereas 5 patient did not show clinical benefit from the intervention. Importantly, patients with CR included subtypes expected to show venetoclax vulnerability, such as poor-prognosis ALL with TCF::HLF fusion. We believe BH3-mimetics have clinical activity in children and should be available to pae-diatric haemato-oncology practitioners in well-selected applications.

## 1. Introduction

Venetoclax (ABT-199, BCL2 inhibitor), the best known and trialled BH-3 mimetic, is a small-molecule proapoptotic drug with significant clinical activity, that is considered practice-changing for several adult cancers. Venetoclax, in combination with other drugs, became a standard of care in certain situations in chronic lymphocytic leukaemia (CLL), is increasingly used in multiple myeloma, acute myeloid leukaemia (AML) and myelodysplastic syndrome (MDS), and has been experimentally administered in other malignancies including ALL [1,2,3,4,5,6,7,8,9]. Although in AML, venetoclax is indicated especially in unfit or elderly adults who would likely not tolerate high-dose chemotherapy, it is frequently used in the relapse or salvage setting, and in the future may be a component of first line therapies [3,7,9,10,11]. In Poland, venetoclax is not registered for any paediatric applications. In adults, registered applications include: combination with hypomethylating agents for patients with AML disqualified from intensive chemotherapy, in combination with obinutuzumab for previously untreated CLL, in combination with rituximab for patients with CLL treated with at least one prior line of therapy, as a monotherapy for patients with CLL and deletion in 17p region or TP53 mutation when treatment with inhibitor of B-cell receptor pathway is inappropriate or ineffective, and as a monotherapy for patients with or without a deletion in the 17p region or TP53 mutation when treatment with both B-cell receptor pathway inhibitors and immunotherapy is ineffective (https://www.ema.europa.eu/en/documents/product-information/venclyxto-epar-product-information_pl.pdf, accessed on 10 April 2023).

In paediatrics, the tested applications are much fewer but new and exciting results were recently presented establishing venetoclax as an active agent in relapsed or refractory leukaemias, and strongly supporting clinical attempts or trials of BH3 mimetics in other diseases and clinical scenarios in childhood cancers [2,12,13,14,15,16,17,18,19]. The importance of the paediatric-relevant experience in AML/MDS, which are main areas of development in adults, is linked to its relatively low prevalence and differences from older patients in terms of genetic features and heterogeneity of myeloid neoplasia in children, resulting in difficulties in extrapolating data on BH3 mimetics from adults.

Thus far, the most important publication on venetoclax use in paediatric oncology is the work by Karol et al. reporting the NCT03194932 trial, i.e., a phase 1 dose-escalation study of Venetoclax in combination with chemotherapy in paediatric patients with refractory or relapsed AML. A total of 36 patients received 28-day venetoclax cycles at either 240 mg/m^2^ or 360 mg/m^2^, in combination with cytarabine, with or without intravenous idarubicin. The recommended phase 2 dose of venetoclax was found to be 360 mg/m^2^ combined with cytarabine, with or without idarubicin. Overall responses were observed in 69% of the 35 patients who were evaluable after the first cycle. Among the 20 patients treated at the recommended phase 2 dose, an impressive 70% complete response rate was reported. The authors concluded that venetoclax in combination with chemotherapy demonstrated excellent activity in this heavily pretreated paediatric population, and compared favourably to results from previous trials in relapsed AML in children [13]. The study provides a strong rationale for venetoclax use in children with relapsed or refractory AML, and supports studies in less heavily pretreated populations including as a front-line treatment.

Continuing the AML setting, another milestone paediatric and adolescent and young adult (AYA) study of venetoclax is a retrospective report from Children’s Hospital Colorado by Winters et al. on the use of a venetoclax/azacitidine combination in six patients with AML and two with MDS. All four AML responders achieved minimal residual disease (MRD) negativity and three of them proceeded to HSCT [19]. Venetoclax was also shown to be a useful component of HSCT-bridging therapies in children with AML [14].

A study by Pullarkat et al. reported on a phase I dose-escalation NCT03181126 study that combined venetoclax with low-dose navitoclax, a BCL-XL/BCL2 dual inhibitor in patients with relapsed/refractory ALL or lymphoblastic lymphoma. The study population included 12 children and the overall response rate in this subgroup was 75% (9/12), and 6 out of those (67%) achieved MRD negativity. Seven of the twelve paediatric patients proceeded to HSCT or CAR-T therapy. This study is significant not only because of high response rates but also because of the novel, somewhat surprising approach, in which two BH3 mimetics were combined [17].

Consistent with pre-clinical data for T-ALL [20,21,22,23], there are case descriptions or case series in T-ALL and T-NHL, with a few paediatric reports within mainly data from adults. Among those, the most significant paediatric data are a report by Gibson et al. on venetoclax use in ALL/LL in 18 children/AYA, 13 of which had a T-cell disease. The authors reported that 5/6 T-cell LBL and 5/7 T-ALL achieved complete remission with or without residual neutropenia or thrombocytopenia (CR/CRi) after a median of one venetoclax cycle combined with nelarabine or decitabine, and 7 of them remained alive within a median follow-up time of 12.11 months [2]. Further, venetoclax in combination with decitabine was presented as a novel salvage therapy for relapsed/refractory paediatric T-NHL [12].

BH3 mimetics, and in particular venetoclax, were also highlighted as potential interventions to target specific vulnerabilities in some aggressive molecular subtypes of ALL, including hypodiploid, *KMT2A*-rearranged or ALL with *TCF3::HLF* fusion, or early T-precursor (ETP)-ALL that are associated with poor or very poor outcomes. With respect to *TCF3::HLF* subtype venetoclax vulnerability, while striking, this is only a correlative link without mechanistic explanation, whereas KMT2A rearrangements seem to be due to epigenetic regulation of the expression of BCL2 via histone modifications [21,24,25,26,27,28,29].

Finally, there are pre-clinical data in neuroblastoma revealing specific *MYCN*-amplified neuroblastoma sensitivity to BH3 mimetics [30,31,32]. A few further studies reported on promising venetoclax-based combinations in neuroblastoma [32,33,34]. Clinical studies of recurrent paediatric solid tumours including neuroblastoma were initiated [34].

Although currently, venetoclax has not been incorporated into paediatric treatment schedules in Poland, it came to our attention that this drug has been already used in several patients that failed conventional therapy in a few centres. The direct aim of the study was to gather the clinical data of all paediatric patients treated thus far with venetoclax in Poland, especially to identify the triggers for the interventions, preferred schedules and potential response markers. We set out to gather this experience of the use of venetoclax in Polish paediatric haemato-oncology departments to potentially support choosing the right clinical context for the drug, planning treatment schedules and alleviating practical problems thus translating into patient benefits and stimulating further research.

## 2. Materials and Methods

### 2.1. Questionnaire

The questionnaire regarding the use of venetoclax was sent to all 18 Polish paediatric haemato-oncology centres. We asked our collaborators to provide anonymized data on all patients aged 0 to 18 years old who received venetoclax for any malignant disease since 2015. There were no exclusion criteria for the survey. Typically, around 1000–1200 children are diagnosed each year in Poland with malignancies and treated in the 18 centres. The gathered anonymised data included: immunophenotypic and molecular data, treatment protocols with risk stratification, disease status, inherited disorders, comorbidities, indications for venetoclax use, cytology, cytometry, PCR-MRD (if applicable), dose of venetoclax, monotherapy or combination therapy, scheme, source for scheme, specific side effects, acute tumour lysis syndrome (y/n), HSCT (y/n), remission (y/n), death (y/n), cause of death and whether any prognostic markers for response were explored.

The Local Ethical Committee at Medical University of Lodz issued a decision stating that the study does not have features of an experiment on human beings and approved the proposed anonymized survey (decision No. RNN/48/23/KE). These data as of 15 December 2022 were gathered and analysed for this report.

The immunophenotypic and genomic data were derived from patient medical records and national registries. As they are database-derived and generated as a part of standard treatment protocols in certified reference regional or national diagnostic laboratories they are not presented here apart with the exception of providing the exact protocol descriptions used for the genomic studies presented in Figure 1 (below).

### 2.2. Response Definitions

We assessed reaction to venetoclax using the AML-BFM (Acute Myeloid Leukaemia Berlin–Frankfurt–Munster) protocols: complete haematological remission (CR) was defined as <5% blasts in the bone marrow without signs of extramedullary disease and complete morphologic count recovery (neutrophils ≥ 1000/µL and platelets ≥ 100,000/µL, independent of transfusion); or aplasia without regeneration (NEL, no evidence of leukaemia) described according to the AML-BFM study group as no detectable malignant blasts (<5%) in BM or peripheral blood; no extramedullary blasts without signs of haematopoietic regeneration and achievement of at least a partial regeneration of the blood thrombocytes <50,000/µL, neutrophil granulocytes < 500/µL and leucocytes < 1000/µL; or lack of response (NR)—not fulfilling the criteria above.

There was one patient with JMML in whom the above criteria were not used and instead we referred to proposed JMML-specific criteria, in which complete clinical remission (cCR) requires all six criteria listed below: white blood cell count below 15 × 10^6^/µL, less than 1% myeloid and erythroid precursors in bone marrow, platelets above 100 × 10^3^/µL, less than 5% blasts in bone marrow, no splenomegaly and no extramedullary disease for at least 4 weeks [35].

### 2.3. Genomic Studies (for Patient 1 Presented in Text and Figure 1)

#### 2.3.1. Microarray Analysis

Copy number variation analysis was performed using a CytoScan HD array (Applied Biosystems, Thermo Fisher Scientific, Waltham, MA) and Chromosome Analysis Suite v 4.2 software (ChAS, Thermo Fisher Scientific, Waltham, MA) as described previously [36]. The experimental data were analysed in two categories: genome-wide CNAs; 5 Mbp and leukaemia-associated region/gene-specific CNAs (leukaemia genes_all_20150505; Fullerton Overlap Map_hg19).

#### 2.3.2. RNA Sequencing

RNA sequencing was performed using the TruSight RNA Pan-Cancer panel (Illumina, San Diego, CA) which contains 1385 cancer genes and enables fusion calling and variant detection within the panel. Twenty nanograms of RNA was processed according to the manufacturer’s protocol and was sequenced on a Next Seq 550 system (Illumina, San Diego, CA) using the NextSeq^®^ Reagent Kit v3 (150 cycles) with a PE NextSeq^®^ Flow Cell. Data analyses were performed using the Illumina BaseSpace apps TopHat Alignment (version 1.0.0, read mapping on hg19 reference genome by TopHat21, fusion calling by TopHat-Fusion2) and RNA-seq Alignment (version 1.1.0, read mapping on hg19 reference genome by STAR3, fusion calling by Manta4 using standard settings) (https://basespace.illumina.com/apps). Fusion transcripts with a low number of split-reads were excluded as likely false positives. The raw data of the sequence variants were converted to variant call format (vcf) files and analysed in Variant Studio software v.4.0.

#### 2.3.3. DepMap Cancer Dependency Map Portal Data Access

We accessed the public version of the DepMap Cancer Dependency Map portal at https://depmap.org/as provided by Broad Institute (as accessed on 5 March 2023) to prepare a list of cancer cell lines most sensitive to *BCL2* loss in a genome-wide CRISPR loss-of-function screen. The data for venetoclax were not used as haematopoietic cancer cell lines were not available in the public version of the compound screen.

## 3. Results

### Clinicopathological and Genetic Characteristics of Patient Cohort

We received responses from 11 centres, 5 of which administered venetoclax to their patients. Table 1 presents the relevant demographic, clinical, genetic and response information on the patients that received venetoclax treatment. Ten children (median age of 8, range 2–17 years), three girls and seven boys, were reported during the data collection. All patients that received venetoclax were treated for haematological cancers: AML (n = 4, including 1 ML-DS), ALL (n = 3), MDS (n = 1), JMML (n = 1) and MPAL (n = 1).

Patient 1 was a pre-school child with previously unrecognised neurofibromatosis type 1 (subsequently a novel p.Val744AlafsTer3 *NF1* variant was revealed). The patient was admitted to the hospital with a WBC count above 200,000/µL and organomegaly. AML M1 was diagnosed and the treatment according to AML BFM 2019 was initiated, and the genetic work-out demonstrated chromosome 7 monosomy and loss-of-heterozygosity (LOH) of the whole q arm of chromosome 17 involving the *NF1* gene. The child demonstrated no blast reduction during a short multidrug prophase and no response to two induction chemotherapy cycles (HAE, HAM). As a salvage therapy, he was scheduled for a 28-day course of venetoclax (360 mg/m^2^) with concurrent IDA-FLA chemotherapy since day 8. His bone marrow blast count on day 1 of this intervention was 55% and dropped to 2% on day 8 (after 7 days of venetoclax monotherapy). FC-MRD on recovery after IDA-FLA/venetoclax was 0.2% with normal haematological values and the patient proceeded to unrelated donor HSCT. Initially, post HSCT, there was a full donor-chimerism, but after 4 months there was an autologous signal and subsequently overt relapse (8% blasts) despite cessation of immunosuppression. The patient was re-challenged with venetoclax but after two weeks the blast count increased to 20% and the patient deteriorated clinically. Lack of response might have been partly due to intestinal GvHD and poor absorption. The child received a salvage treatment with a combination of azacitidine and gemtuzumab ozogamicin and finally succumbed to the disease. No further genomic variants potentially relevant to venetoclax response or significant for AML diagnosis and treatment were found using RNA-seq. Figure 1A demonstrates that chromosome 7 monosomy is a high-risk disease marker as well as LOH of chromosome 17q revealed in microarray analysis of the leukaemic genome.

There was a variety of cancer-driving germline and somatic variants and genomic lesions revealed in the cohort. In the ALL and MPLA settings, these included germline *TP53* variants as well as hypodiploidy and recurrent gene fusions including *TCF3::HLF* and *SET::NUP214*. In AML, the reported genomic findings included germline 21 trisomy, germline *NF1* and *RUNX1* variants and somatic *GATA1* variant as well as *KMT2A* rearrangements. In patients with JMML, somatic *NRAS* mutations were revealed. It should be noted that this is a subgroup of patients with various forms of treatment failures so it is likely enriched for unfavourable genetics. A high (4/10) proportion of patients had congenital genetic syndrome. The patients included in this report were heavily pretreated with 1–5 prior lines of therapy, with 3/10 having undergone allo-HSCT as one of previous lines of therapy.

The children were administered film-coated tablets. One patient that could not swallow tablets received crushed tablets suspended in 5% dextrose and while the bioavailability of such formulation is not known, that patient had an immediate, undoubtful response. A paediatric solution is not currently available in Poland. In 8/10 patients, the dose was 360 mg/m^2^; one patient with ML-DS was administered 240 mg/m^2^ and in one case the dose was escalated from 20 to 200 mg/m^2^. A total of 9/10 patients received venetoclax with concomitant chemotherapy (as specified in Table 1).

In general, clinical benefit (CR, cCR—complete clinical response according to JMML criteria or MRD improvement bridging to HSCT in a patient without overt relapse) was reported in five patients out of ten. No clinical benefit was observed in the remaining five patients. Specifically, there were 4/10 CRs observed which proceeded to HSCT/tisagenlecleucel; however, one of the patients was in complete haematological remission before the intervention that was triggered by positive FC-MRD post previous negative FC-MRD (and FC-MRD was again negative after intervention). In the patient with JMML, venetoclax was used as a bridge monotherapy before HSCT which decreased leucocyte count, normalised the platelet count and improved the organomegaly, consistent with cCR (clinical complete remission according to JMML criteria) [35]; however, these results were from a short observation as the patient proceeded to HSCT so the term must be used cautiously.

Four of the ten patients who entered venetoclax treatment without morphological recovery and with a very high comorbidity status displayed aplasia without regeneration and died because of infectious complications in aplasia with an unclear disease status. One patient (MPAL) did not present any response after two cycles of venetoclax and was switched to other therapies.

None of the reported complications/co-morbidities could be clearly linked to venetoclax in this heavily pretreated patient population, and frequently had active neoplastic diseases. Further, as this was a retrospective/survey study, there was no uniform manner of gathering toxicity data. No serious adverse event reports were mentioned in the patient medical data. Consequently, we decided not to report on toxicities as this would be misleading and difficult to interpret. ALL data are summarised in Table 1 and patients with a favourable response or the most significant clinical scenarios are presented in detail below.

Patient 2 with a *TCF3::HLF*-positive B-cell precursor acute lymphoblastic leukaemia did not achieve remission following induction and HR1 chemotherapy as per AIEOP-BFM 2017 (PCR MRD 4 × 10^−2^) and after the first blinatumomab cycle, progression was noted (PCR MRD rising to 1 × 10^−1^). As a salvage therapy, the child was subsequently administered HR2 and HR3 AIEOP-BFM cycles with concurrent escalating-dose venetoclax. This resulted in negative PCR-MRD after two cycles of the HR chemotherapy (HR2, HR3)/venetoclax combination. The second HR cycle was complicated with hyperbilirubinemia and pancytopenia causing temporary withholding of venetoclax. The patient subsequently underwent autologous CAR-T therapy (tisagenlecleucel) and remained in remission for 6 months until the routine bone marrow biopsy demonstrated relapse with a PCR-MRD of 3 × 10^−2^ and 9.7% lymphoblasts of the initial immunophenotype in the FC-MRD. The patient was treated with one cycle of INN (international non-proprietary name) inotuzumab ozogamicin, proceeded to allo-HSCT and has remained in remission for 13 months until now. Figure 1B demonstrates the gain of exon 4 of the *HLF* gene which is involved in the unbalanced translocation with the TCF3 fusion partner. The *TCF3::HLF* gene fusion was confirmed by FISH and RNA sequencing (and no disease or venetoclax-response relevant sequence variants were found in RNA-seq).

Patient 3 with Li-Fraumeni syndrome, previously treated for choroid plexus carcinoma, was diagnosed with T-ALL with mediastinal tumour. The patient achieved haematological remission on D33 of treatment according to AIEOP-BFM 2019, but showed molecular relapse (increasing PCR-MRD to the level of 3 × 10^−3^) post HR1 block. As a high BCL2 expression was earlier revealed in the lymph node immunohistochemistry, venetoclax was considered as a salvage intervention. Eventually, venetoclax was implemented along with the second line treatment as per the IntreALL protocol and was continued during third line therapy (nelarabine/cyclophosphamide/etoposide followed by nelarabine/topotecan/vinorelbine/thiotepa) and up to HSCT conditioning with good tolerance and moderate PCR-MRD reduction. The child underwent HSCT complicated with acute intestinal and cutaneous GvHD and received venetoclax for one year. PCR-MRD was negative on day 100 post HSCT and remained negative 6 months after transplantation. Figure 1C demonstrates a 17p13.1–p13.3 deletion encompassing the *TP53* gene in the leukaemic genome that was identified using microarray analysis.

Patient 4 was diagnosed with AML with a *KMT2A::MLLT3* fusion two years after treatment due to Wilms tumour stage III, and was qualified for allo-HSCT as a consequence of no remission at day 28 of induction. A negative FC-MRD after the second induction was achieved. Although the VP-16/ARA-C consolidation cycle was used to control the disease during the prolonged pre-transplant period, an FC-MRD of 0.3% was detected and the patient was treated with venetoclax and azacitidine until a negative MRD was obtained. The patient remains in remission after the subsequent allo-HSCT.

Patient 5, diagnosed with JMML after two allotransplantations, was treated with one cycle of venetoclax during his second relapse after ineffective azacitidine treatment (increasing leucocytosis). The therapy was considered successful due to normalization of haematological values and clinical and radiological improvement. The patient underwent allo-HSCT from an unrelated donor complicated with cutaneous GvHd and EBV reactivation and has remained in remission for 7 months until now.

In patient 6, venetoclax was implemented due to the second relapse of mixed-phenotype acute leukaemia (myelo/T MPAL) after failing to achieve remission with nelarabine/ARA-C. There was no response to venetoclax even after adding carfilzomib (NR). The patient was switched to another therapy.

Patient 7 diagnosed with ML-DS associated with a *GATA1* mutation was treated ineffectively with venetoclax/AZA as a salvage therapy during his early relapse complicated with a COVID-19 infection in view of previous very severe adverse effects of chemotherapy. A bone marrow cytology after 1 month was classified as NR.

Patients 8, 9 and 10 were heavily pretreated individuals who were administrated venetoclax with concurrent chemotherapy as a salvage therapy for relapsed leukaemia that failed to respond to one or more induction schemes. In patients 9 and 10, treatment was discontinued (after 19 and 18 days, respectively) due to severe pancytopenia and infections, and the cytology studies were consistent with NR (no response) while patient 8 received a full scheduled cycle and a biopsy revealed aplastic bone marrow in the context of no haematological recovery. The patient gradually deteriorated and died with an unclear disease status.

#### DepMap Cancer Dependency Map Portal Data Access of BCL2 Cell Line Sensitivity

We accessed the DepMap Cancer Dependency Map portal at https://depmap.org/as provided by the Broad Institute (on 5 March 2023) to generate a list of cancer cell lines that are the most sensitive to BCL2 loss in a genome-wide CRISPR loss-of-function screen. The list of the top 20 BCL2-sensitive cell lines is provided in Appendix A. A high proportion (8 of 20) are paediatric cell lines (BCP-ALL, AML, neuroblastoma). 

## 4. Discussion

Our multi-centre study of venetoclax use has revealed clinical activity in children with diverse types of blood malignancies. We recorded that venetoclax with or without chemotherapy could achieve (or at least consolidate) CR or cCR (JMML) in 5/10 patients with poor prognoses and venetoclax served as a bridge therapy before allo-HSCT or CAR-T therapy. This response rate is lower than the impressive 70% CR reported by Karol et al. in the NCT03194932 trial among patients treated at the recommended phase 2 dose or the 67% MRD negativity reported by Pullarkat et al. in the trial. This likely reflects that fact that the several patients included in our survey would not qualify for clinical trials and venetoclax was a ‘last resort’ intervention.

While our retrospective survey-based report on venetoclax use in Polish haemato-oncology centres does not allow us to draw conclusions on the rate of response to venetoclax in paediatric malignancies, it demonstrates that BH3 mimetics may show considerable clinical activity in a setting of salvage treatment of resistant, heavily pretreated disease. It is reasonable to speculate that response rates would be higher outside ‘salvage’ situations, which are associated with therapy resistance, poor patient performance and side effects/toxicities of previous treatments, preventing the chances for response and patient survival.

A significant observation in our cohort was that each patient that achieved CR had a very specific molecular subtype of the malignancy or a cancer predisposition syndrome and while there were no obvious mechanistic links to BH3 mimetic responses, this allows us to suggest that it is worth establishing which molecular subtypes of paediatric malignancies might display specific venetoclax vulnerability. This is consistent with efforts from several pre-clinical studies that highlighted genetic backgrounds associated with high BH3-dependance and BH3 mimetic sensitivity. Thus far, in the setting of ALL and to a lesser extent in AML, the pre-clinical experimental data have pointed to several molecular subtypes which may display therapeutically applicable venetoclax vulnerabilities [21,24,25,26,27,28,29].

These especially include ALL with *KMT2A* translocation, ALL with *TCF::HLF* fusion, early precursor T-ALL (ETP-ALL) and T-ALL with *RPL10* mutations. Specifically, high expression of BCL2-family proteins was seen in *MLL*-rearranged and hypodiploid BCP-ALL. KMT2A::AF4 directly regulates *BCL2* gene expression by promoting increased H3K79me2/3 levels in the *BCL2* gene vicinity leading to a BH3-dependance in *KMT2A*-rearranged leukaemia [24]. Other pre-clinical reports also point to venetoclax activity in *KMT2A*-rearranged ALL [27,29]. The same study also demonstrated high BCL2 expression in hypodiploid BCP-ALL but the consequences were not studied in more detail [24]. There is, however, another pre-clinical study that demonstrated venetoclax activity in hypodiploid ALL [26]. An approach combining extensive genomic analysis and drug sensitivity profiling of *TCF::HLF*-positive ALL led to a demonstration of an exquisite sensitivity of *TCF3::HLF*-positive ALL to venetoclax and the conclusion that this could be a relevant druggable dependency [28]. BCL2 also seems to be specifically upregulated in ETP-ALL and T-ALL with *RPL10* mutations and that corresponded to venetoclax sensitivity [37,38]. Apart from those backgrounds, there are relatively many reports on venetoclax clinical activity in T-ALL (mainly case series), but it is difficult to conclude whether this represents a more general sensitivity in T-ALL or if this is a bias of clinical decisions or reporting. However, many authors believe that venetoclax should be considered in recurrent or refractory T-ALL [2,21,37,38,39,40,41]. It is not known if there are common genomic/molecular features of those responsive T-ALL cases, apart from those mentioned above, and from the pre-clinical data on ETP-ALL and *RPL10*-mutated T-ALL.

There are no data specific for paediatric AML regarding markers of venetoclax response, whereas data from studies on adult populations frequently associated it to genetic variants/lesions that are not prevalent and applicable in children. These data describe patients’ populations with higher or lower response rates rather than sharply defined groups with venetoclax vulnerability [42,43,44]. As they mostly refer to markers that are not frequently identified or actionable in paediatric AML, there is a paucity of data on paediatric AML/MDS and venetoclax response; despite this fact, there are more established venetoclax applications in AML than in ALL.

Consistent with the literature presented above, the patient with *TCF::HLF*-positive BCP-ALL with refractory disease achieved CR using the chemotherapy/venetoclax combinations and proceeded to CAR-T therapy (patient 2). Further, the patient with *KMT2A*-rearranged AML achieved FC-MRD negativity following a venetoclax course pre HSCT (FC-MRD 0.3% before venetoclax treatment), which may correspond to venetoclax activity. Consistently, in our DepMap data access, two paediatric *KMT2A*-rearranged cell lines were among the top 20 *BCL2*-dependent cell lines. Additionally, there was a CR in a patient with T-ALL and Li-Fraumeni syndrome (patient 3). Although there are no data on venetoclax response in a *TP53*-mutated background (germline or somatic), it is of note that venetoclax-sensitive hypodiploid BCP-ALL is one of the possible presentations of Li-Fraumeni syndrome and *TP53* is very frequently mutated in hypodiploid ALL. Similarly, there is a report on venetoclax responses in *TP53*-mutant T-ALL [41].

The patient with the most spectacular response was a child with *NF1* and refractory AML (patient 1). There are no obvious data linking *NF1* aberrations, either somatic or germline, with venetoclax response. It is of note, however, that in the paediatric trial NCT03194932, all three AML patients with somatic *NF1* mutations demonstrated BCL2 dependence in BH3-dependence testing and two of them demonstrated CR with a FC-MRD < 0.1% post first cycle, while the third PR had an FC-MRD between 0.1 and 5% [13].

To our knowledge, there are no data to speculate on the molecular mechanisms or associations of the observed venetoclax response in JMML.

There are several limitations to our study: it was conducted retrospectively, the patient sample is small and heterogenous, and there were many confounding factors and markers of potential response (BH3-dependence or even BCL2 expression) that were generally not studied. In particular, in some patients, the intervention was initiated with very advanced disease and comorbidity load and thus in that setting a clinical benefit is very unlikely to be seen. Further, given the small numbers of patients in the potential subgroups, there was no room for meaningful statistical analysis. Nevertheless, even with all these limitations that decreased the power of the study, we could clearly observe patients with an obvious response to the drug and some of this strongly agreed with predicted venetoclax vulnerabilities.

## 5. Conclusions

The current status of venetoclax use in Poland in paediatric population is thus far not based on any national consensus and is triggered by individual patient’s need for salvage non-chemotherapy-based interventions rather than firmly predicted response. The drug has not been used in Poland outside of leukaemia settings. Still, there are a few encouraging conclusions from our survey report: BH3 mimetics should be further studied in paediatrics in both pre-clinical models and in clinical trials. There are likely specific subgroups of patients that may significantly benefit from this treatment. These subgroups/markers should be delineated in pre-clinical and clinical markers/ex vivo sensitivity studies and carried on to clinical trials or at least used for sub-analysis of unselected patient populations. From a practical view, venetoclax and similar compounds should be readily available in paediatric oncology centres. Based on pre-clinical and clinical data as well as our preliminary observations, this intervention should be especially considered in, but not limited to, salvage therapies for refractory ALL with *TCF3-HLF* fusion or *KMT2A* rearrangements and any refractory AML. A decision regarding this intervention could be potentially facilitated in the future by deeper investigations of markers of potential venetoclax response.

## Figures and Tables

**Figure 1 children-10-00745-f001:**
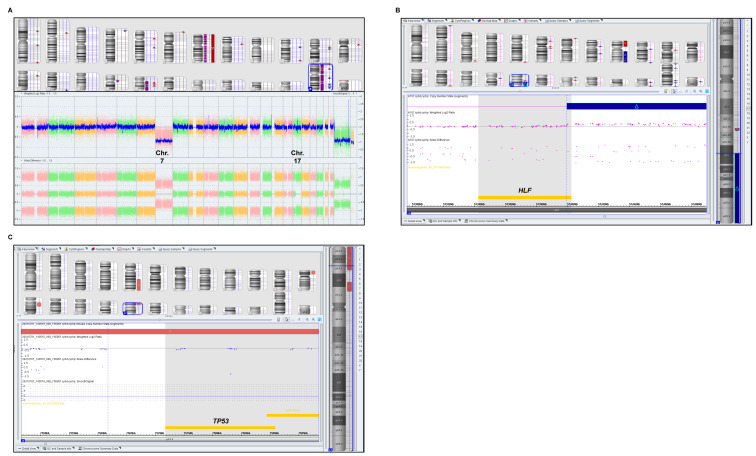
Results of single nucleotide polymorphism array analysis of the leukaemic genomes in selected patients treated with venetoclax. (**A**) Whole genome view showing monosomy 7 and LOH of chromosome 17q in patient 1. (**B**) Chromosomal microarray analysis data plot for chromosome 17 displaying the gain of 17q22–q25.3, which involves exon 4 of the HLF gene in patient 2. (**C**) CMA data plot showing the 17p13.1–p13.3 deletion encompassing the TP53 gene in patient 3. Copy number abnormalities were identified by a decrease (deletion) or an increase (gain) in the Log2 ratio. The coordinates are in reference to GRCh37/hg19. 1.

**Table 1 children-10-00745-t001:** Characteristic of the study group.

No/Age Category	Diagnosis	Congenital Syndrome	Germline Variant	Somatic Genetics	Disease Status	Response	Dosing Scheme	Concurrent Therapy	HSCT or CAR-T Post Venetoclax	Relapse Post Venetoclax	Death Post Venetoclax	OS (Months)
1/PS	AML with MDS-related cytogenetic abnormalities	*NF1*	*NF1* (NM_001042492.3):c.2226_2229del (p.Val744AlafsTer3)	Monosomy 7 * LOH of chromosome 17q involving *NF1*	Primary refractoriness	CR	360 mg/m^2^ for 28 days (1 cycle)	IDA-FLA according to the AML-BFM 2019	allo-HSCT	yes	yes	7
2 SA	BCP-ALL	no		*TCF3::* *HLF* fusion gene *	Primary refractoriness	CR	Escalating every 7 days in 1st cycle 20–50–100 mg/m^2^, 2nd cycle 50–100–200 mg/m^2^ ** (2 cycles)	HR2 (1st cycle) and HR3 (2nd cycle) chemotherapy according to AIEOP-BFM 2017	CAR-T	yes	no	25
3/PS	T-ALL (post choroid plexus carcinoma)	Li-Fraumeni	*TP53* (NM_000546.5):c.469G > T (p.Val157Phe)	Deletion of chromosome 17p involving *TP53 **	Molecular relapse (increasing PCR-MRD)	CR ***	400 mg/m^2^ during II and III line treatment until conditioning and from day 28. post HSCT	Various chemotherapy schemes (HIB according to IntreALL, TVTC, nelarabine) and allo-HSCT ****	allo-HSCT	no	no	12
4/SA	AML with KMT2A::MLLT3 fusion (post Wilms tumour)	no		t(9;11) (p22;q23); *KMT2A::MLLT3*	CR, positive FC-MRD post previous negative result	CR with negative FC-MRD	360 mg/m^2^ for 28 days (1 cycle)	ARA-C	allo-HSCT	no	no	6
5/PS	JMML	no		*NRAS* mutation	2nd relapse post 2 allo-SCT	cCR	360 mg/m^2^ for 28 days (1 cycle)	-	allo-HSCT	no	no	7
6/PS	MPAL, T/myeloid	no		*SET::* *NUP214* fusion gene	2nd relapse post 2 allo-HSCT	NR	360 mg/m^2^ for 28 days (2 cycles)	ARA-C, carfilzomib, azacitidine	no	-	no	8
7/PS	ML-DS	Down syndrome	Trisomy 21	*GATA1* mutation	1st relapse	NR	240 mg/m^2^ for 28 days (1 cycle)	ARA-C, azacitidine	no	-	yes	2
8/SA	AML NOS	no		*KMT2A::MLLT1*	1st relapse	NEL	360 mg/m^2^ for 28 days (1 cycle) **	Azacitidine	no	-	yes	1
9/SA	AML with myelodysplasia-related gene mutations	Familial platelet disorder	*RUNX1* (c.806-2A > C)		2nd relapse post 2 allo-SCT	NR	360 mg/m^2^ for 28 days (1 cycle) **	IDA+ ARA-C	no	-	yes	1
10/SA	BCP-ALL	no		monosomy 12, hypodiploidy	1st relapse	NR	360 mg/m^2^ for 28 days (1 cycle) **	HC1 according to the IntreALL 2010	no	-	yes	2

Abbreviations: IN—infant (age < 12 months), PS—preschool (age ≤ 6 years), SA—school and adolescent (age 7–18 years), AIEOP-BFM—Associazone Italiana Ematologia Oncologia Pediatrica (AIEOP) and the Berlin–Frankfurt–Münster (BFM) group, AML—acute myeloid leukaemia, BCP-ALL—B cell progenitor acute lymphoblastic leukaemia, T-ALL—T cell acute lymphoblastic leukaemia, JMML—juvenile myelomonocytic leukaemia, MPAL—mixed-phenotype acute leukaemia, NOS—not otherwise specified, *NF1*—neurofibromatosis type 1, LOH—loss of heterozygosity, FC-MRD—flow cytometry minimal residual disease, PCR-MRD—polymerase chain reaction minimal residual disease, CR—complete haematological remission, cCR—complete clinical response according to JMML-specific criteria, NEL—aplasia without regeneration (‘no evidence of leukaemia’), OS—overall survival (months), *—genetic investigations discussed in the manuscript, **—or until discontinuation due to poor clinical condition, ***—achieved post HSCT, ****—continues venetoclax monotherapy as maintenance post HSCT. Explanation of chemotherapy cycle names: IDA-FLA (fludarabine, cytarabine, idarubicin), ARA-C—cytarabine, VP-16—etoposide, HR1/HR2—‘high-risk’ 1 and 2 chemotherapy cycles according to AIEOP-BFM 2017.

## Data Availability

All other data generated during the study are available from the corresponding author upon reasonable request.

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
