# Peer review of "Venetoclax Use in Paediatric Haemato-Oncology Centres in Poland: A 2022 Survey"

_children, 2023, doi:10.3390/children10040745_

Round 1

Reviewer 1 Report

In general, I do not have any specific objections regarding the paper. It is well-written, the data is presented clearly and the results are supported by relevant literature. However, I have a huge concern regarding the lack of patients' informed consent (or patients' parents since all patients are children). The paper describes in details clinical data about each patient in whom venetoclax was administred, and their identity could be easily identified. This inevitably requires informed consent and institutional ethical approval, regardless of the fact that the design of the study is a survey.

Author Response

We would like to express our sincere gratitude for the reviewers and editors for their time and effort devoted to providing insights aimed at improving our manuscript.

We provide a revised version of the manuscript and a detailed response to reviewers’ requests.

Reviewer 1

The paper describes in details clinical data about each patient in whom venetoclax was administred, and their identity could be easily identified.

We address the concern of the reviewer by removing data that could lead to patient identification. While we received approval for our anonymised survey from the Local Ethical Committee at Medical University of Lodz we acknowledge Reviewer 1 and 2 views that some of the data are unnecessary (their removal does not impact the message), could potentially lead to patient identification and should be best removed to clear ethical concerns.

In the revised manuscript we remove the age (only coding now as infant /pre-school /school and adolescent age) and sex of the patients.

In Poland there are annually circa 250 newly diagnosed children with acute leukemia. Over the survey period of around 6 years this makes approximately 1500 patients. We believe that currently this survey and report are truly anonymous (no information on centre, sex, age).

If there are any other data considered by the reviewers or editors as sensitive we are of course ready to remove them or code differently.  

We add the details of the Ethical Committee approval.

Reviewer 2 Report

This paper evaluates the venetoclax use in pediatric patients in Poland, and the main contribution is related to case descriptions enrolled in the survey. Venetoclax seems to be a promising drug for acute myeloid leukemia (AML) therapy, with the recent approval by the FDA for the treatment of elderly patients and younger unsuitable for induction chemotherapy. This new drug, an inhibitor of BCL2, is also being tested in clinical trials and real-world experience, with optimistic results, inclusive in the first-line therapy. However, little is known about pediatric patients, besides some case reports (Gibson et al., 2022) that are published. Therefore, this survey and analysis will bring important information to the literature.

Although this research brings valuable information, there is a major concern about the conduction of evolving human beings, such as the questionnaires, and sample analysis.  According to the Declaration of Helsinki, it is necessary the mention of ethics committee approval in the material and methods section. Please revise and add this substantial information to the paper.

In the abstract and in the introduction, it is not clear what is the aim of this research. I understand that this work started with a survey, and it was an important step to achieve the results. However, many analyses and case reports were described, and this specific aim explaining the purpose must be emphasized.

In the introduction, I recommend adding a paragraph with epidemiological data about AML prevalence, and incidence, especially in the pediatric context to get the emphasis on the rarity, severity, and heterogeneity of this entity. We also recommend the use of the new International Consensus Classification (ICC), which includes AML (Arber et al Blood 140 (11): 1200-1228 doi: 10.1182/blood.2022015850.).

The title also doesn’t express all the information analyzed in this work by the group. It was supposed to give information about the experience of venetoclax use in pediatric centers in Polland. However, the survey was a starting point for a case report description and little consensus is summarized about the use. There is little statistical data analyzed and conclusions drawn, and a lot of patient case report descriptions.

In the methodology section, some information is missing:

Also, more precise information about the number of centers enrolled in this study will bring a clear dimension of how many centers were involved, especially whether correlated with how many centers there are in Poland and how many were enrolled, instead of the use of “all Polish Pediatric onco-hematology centers”. In the results, it is possible to have a proportional dimension of the study as it is described that 11 centers answered the surveys. However, we still don’t know how many centers the survey was sent it. See lines 39 (abstract) and 120 (material and methods).

Please add a section talking about the design of the study and the survey, as at the beginning of the discussion section some of this information is introduced, but much information is missing. Also, there is no description of how big is the population analyzed, the inclusion and exclusion criteria, and so on.

However, we also recommend minor corrections, such as:

Revise English words like “haematooncology” for a more suitable one, such as onco-hematological, haemato-oncology or haemato oncology.

Please, update the reactions to venetoclax to the actual usage: CR (complete remission); Cri (complete remission with incomplete count recovery); no CR/Cri in the abstract and table 1.

Please revise the qualities of the images, as is not possible to read the information in figure B and C in the lower panel.

It is not clear which genomic data were derived from medical records and which studies were performed at the center of the study. It is not clear where the research and genomic studies were performed. Please, specify in the methodology section as well as whether all patients had given the informed consent, and explain the reasons for those who were not possible to get it.

Make this information more straightforward to tell the importance of this sentence: “The data as available on Dec 15, 2022 were gathered and analysed for this report.” Maybe telling the interval in which data was collected will bring more information, instead of the deadline.

 The results description of “DepMap Cancer Dependency Map portal data access of BCL2 cell line sensitivity” section will be more informative about whether more data will be added. This section is very poor and is not adding much information as well as correlate it to the clinical and genomic data collected from the medical records, and whether some in vitro tests were performed.

In the discussion, when the group describes the limitations of the study, they also can deepen this information, enumerating which are the confounding factors and which are the markers of potential response, and what correlation made it possible to conclude which patients were good versus bad responders to venetoclax treatment.

The conclusions are very general and should point out a summary of results answering the questions made at the end of the introduction when they claim that they expected to answer by the survey the right clinical context for venetoclax, which treatment schedules will translate into benefits for patients (lines 114-117). I recommend reviewing this sentence in the introduction to better explain the research aims of this work and answer them in the conclusion

Please, revise all acronyms used in the text to be sure all of them have their meaning defined and that the definition matches the acronym.

For example:

In line 138 and 219  please specify what NEL acronym means as the definition doesn’t match the acronym;

In figure 1, Please define what CMA means (lines 248 and 250).

Specify what HR1 in line 255 and 273; The same for HR2 (table 1, and lines 257, 259) and HR3 (table 1, and lines 258 and 259)

The same for: AIEOP-BFM acronym in table 1 and lines 255, 258, 272; PCR-MRD in table 1 and in lines 124, 256, 259, 263, 273, 279, 281; FC-MRD; INN line 265; JMML; IDA-FLA; VP-16/ARA-C; MPAL

Please correct the acronym MN to MPN in line 322, for the most usage for myeloproliferative neoplasms

Please revise typewriting errors like: line 151 “&gt”; line 158 “&#39”; line 165 “<10”, line 316 “od”

Author Response

We would like to express our sincere gratitude for the reviewers and editors for their time and effort devoted to providing insights aimed at improving our manuscript.

We provide a revised version of the manuscript and a detailed response to reviewers’ requests.

Reviewer 2

According to the Declaration of Helsinki, it is necessary the mention of ethics committee approval in the material and methods section. Please revise and add this substantial information to the paper.

We address the concern of the reviewer by removing data that could lead to patient identification. While we received approval for our anonymised survey from the Local Ethical Committee at Medical University of Lodz we acknowledge Reviewer 1 and 2 views that some of the data are unnecessary (their removal does not impact the message), could potentially lead to patient identification and should be best removed to clear ethical concerns.

In the revised manuscript we remove the age (only coding now as infant /pre-school /school and adolescent age) and sex of the patients.

In Poland there are annually circa 250 newly diagnosed children with acute leukemia. Over the survey period of around 6 years this makes approximately 1500 patients. We believe that currently this survey and report are truly anonymous (no information on centre, sex, age).

If there are any other data considered by the reviewers or editors as sensitive we are of course ready to remove them or code differently.  

We add the details of the Ethical Committee approval.

specific aim explaining the purpose must be emphasized.

We rephrased the abstract and the introduction to emphasize and make clearer the aims of the study.

In the introduction, I recommend adding a paragraph with epidemiological data about AML prevalence, and incidence, especially in the pediatric context to get the emphasis on the rarity, severity, and heterogeneity of this entity.

In the introduction we to certain extent rephrased the information on pediatric AML including stressing relatively low prevalence, and heterogeneity of this disease as well as significant differences versus adult AML. Actually, Reviewer 3 requested shortening the introduction so we tried to rephrase rather than extend the introduction to adjust for requests of both reviewers.

We also recommend the use of the new International Consensus Classification (ICC),

While presenting AML patients we now conform with the 2022 ICC classification.

little consensus is summarized about the use. There is little statistical data analyzed and conclusions drawn

We extended and added new information to the Conclusions paragraph. We argue that the data we gathered does not at this stage lead to more definite conclusions than what we present. Also, as we expected, the cohort is heterogenous that most attempts at statistical /numerical analysis are not plausible. We mention these issues as limitations.

In the methodology section, some information is missing:

more precise information about the number of centers enrolled

add a section talking about the design of the study and the survey,

how big is the population analyzed, the inclusion and exclusion criteria, and so on.

In the methodology section, we now provide information on what number of centres was invited and responded /not-responded to our survey.

We now provide the set-up of the study such as inclusion /exclusion criteria, population description, etc.

We provide interval of the survey.

we also recommend minor corrections

We corrected the inconsistent typographical issues as requested.

Please, update the reactions to venetoclax to the actual usage.

We propose sticking with the response definitions and descriptions we used (Treatment Recommendations AML-BFM 2019, AML-BFM study group, version 03/2019, Chapter 2.1). We feel they well express the situation of the patients. This is also related to the fact that Reviewer 3 suggested using another set of definitions so changing would not be compatible with both requests.

Please revise the qualities of the images

We provide a new version of the figure with improved resolution as separate file apart from the figure embedded in the text.

informed consent

Approach of full anonymity explained above

It is not clear which genomic data were derived from medical records and which studies were performed at the center of the study.

Anonymised genomic data are obtained from the national database shared by the pediatric hemato-oncology centres. They were not performed for this study but as per clinical indications. Their conduct is partly centralized that is the assays are performed in a few most specialized, certified centres for the whole country with e.g. one providing FISH service, the other RNAseq. The information on methods provided is related to the results presented in Figure 1 to correlate with the data presented in Figure 1. This was the patient with a spectacular response and we wanted to cover the patients data more thoroughly. Optionally, it can be moved to figure legend or to a supplement.

The remaining genetic data in the survey was obtained with standard methods as per treatment protocol and this is not elaborated on here. We now explain that fact.

telling the interval in which data was collected will bring more information

It is given now.

The results description of “DepMap Cancer Dependency Map portal data access of BCL2 cell line sensitivity” section will be more informative about whether more data will be added. 

As it is not possible to extend the DepMap-based table /paragraph we leave it do reviewers and editors decision whether this should be preserved, removed or perhaps moved to a supplement.

 In the discussion, when the group describes the limitations of the study, they also can deepen this information, enumerating which are the confounding factors and which are the markers of potential response, and what correlation made it possible to conclude which patients were good versus bad responders to venetoclax treatment.

We modify the discussion and conclusions section to provide more direct conclusions and consequences of the study. We also rephrase the corresponding sentences in introduction.

revise all acronyms used in the text

Acronyms were revised.

Reviewer 3 Report

the paper by Bobeff and colleagues is a retrospective report about venetoclax use in pediatric hematological diseases in Poland. The topic is extremely interesting, being venetoclax a revolutionary approach not yet completely explored in the pediatric setting. Even thought, some methodological issues need to be revised and the presentation of data is not always clear. The paper could benefit from some revisions.

-       I would suggest shortening the introduction and modifying the structure. Lines 52-60: I suggest highlighting the main results in adult setting regarding AML, MDS and ALL that can be potentially translated in pediatric diseases. I would also mention diseases in which venetoclax has been approved so far. Lines 62-102: the revision of pediatric data is interesting but too long and hard to follow. I suggest providing main clinical results in AML setting (also including DOI: 10.1038/s41409-022-01877-2) and the few clinical data about pediatric ALL. Regarding preclinical reports from lines 103 to 110 I suggest further discussing studies about particular genetic subtypes, particularly TCF-HLF and MLL-rearranged ALL.

-       In the methods from line 134 to 145 I suggest better modifying the definition of responses. Did you also consider partial response (PR, e.g. blasts 5-20%), treatment failure (TF, discontinuation due to toxicity), non-response (NR). I would also further explain in the text the response criteria used for JMML.

-       From lines 148 to 172 it is not clear how the genetic analysis was determined? Did you perform this analysis centralizing the samples from every patient included? Or data were collected retrospectively from another database?

-       Results: I would generally not add comments in this section (e.g., “a surprisingly high proportion”) that need to be limited to the discussion

-       I would summarize genetic and cytogenetic alterations of included patients in the text

-       It is also essential to mention in the results more details about the different therapies adopted in combination with venetoclax in the included patients, as this is a very important issue that is still unclear (e.g., azacytidine versus standard chemotherapy)

-       The role of venetoclax combination therapies as bridge to CART cells therapy and HSCT should be reported in the text (eg proportion of patients bridged to HSCT and outcomes after transplant)

-       Lines 208-213: details regarding toxicity and complications related to venetoclax need to be reported 

-        Table 1 lacks some essential information: prior therapies / HSCT; details regarding venetoclax cycles (number of cycles, combined therapies, responses after single cycles), HSCT/CART after venetoclax, relapse after venetoclax vs after CART/HSCT, causes of death

-       The “single cases” description is interesting but maybe too hard to follow. I suggest summarizing data regarding different groups of diseases (e.g. ALL vs AML) with different responses

-       Lines 313-318 and Table 2: this part is interesting but maybe it does not fit with the rest of the paper. It can be maybe only mentioned when discussing responses in different genetic subtypes

-       In the discussion I suggest summarizing the “review” part from lines 352 – 381 and discussing more in details the results of the study. It would be interesting to add a comparison of your results with the pediatric reports  published so far

Author Response

We would like to express our sincere gratitude for the reviewers and editors for their time and effort devoted to providing insights aimed at improving our manuscript.

We provide a revised version of the manuscript and a detailed response to reviewers’ requests.

Reviewer 3

With regard to modifying the introduction:

I would suggest shortening the introduction and modifying the structure. Lines 52-60: I suggest highlighting the main results in adult setting regarding AML, MDS and ALL that can be potentially translated in pediatric diseases. Lines 62-102: the revision of pediatric data is interesting but too long and hard to follow.

We propose keeping much focus on pediatric data in our introduction, this is also in agreement with Reviewer 2, who rather stresses ‘pediatric context’.  We would very much prefer preserving the revision of pediatric data in the manuscript.

I suggest providing main clinical results in AML setting (also including DOI: 10.1038/s41409-022-01877-2)

We now include this missing reference suggested by the reviewer.

I would also mention diseases in which venetoclax has been approved so far.

We enumerate the disease were venetoclax is registered in Poland.

Regarding preclinical reports from lines 103 to 110 I suggest further discussing studies about particular genetic subtypes, particularly TCF-HLF and MLL-rearranged ALL.

With respect to TCF3-HLF subtype –while striking, this link is so far correlative data without mechanistic explanation so we cannot provide much further explanation or discussion of the mechanism. Regarding KMT2A rearrangements we add the information that this is related to KMT2A/MLL-dependent histone modifications and epigenetic regulation of expression of BCL2 in introduction and this is also described in discussion.

Methods:

In the methods from line 134 to 145 I suggest better modifying the definition of responses. Did you also consider partial response (PR, e.g. blasts 5-20%), treatment failure (TF, discontinuation due to toxicity), non-response (NR). I would also further explain in the text the response criteria used for JMML.

We added the explanation of response criteria in JMML as in the paper by Niemeyer.

Apart from JMML, we propose sticking with the response definitions and descriptions we used (Treatment Recommendations AML-BFM 2019, AML-BFM study group, version 03/2019, Chapter 2.1). We feel they well express the situation of the patients. This is also related to the fact that Reviewer 1 suggested using another set of definitions so changing would not be compatible with both requests anyway. We believe too many categories would be overwhelming in 10 patients described.

From lines 148 to 172 it is not clear how the genetic analysis was determined? Did you perform this analysis centralizing the samples from every patient included? Or data were collected retrospectively from another database?

Anonymised genomic data are obtained from the national database shared by the pediatric hemato-oncology centres. They were not performed for this study but as per clinical indications. Their conduct is partly centralized that is the assays are performed in a few most specialized, certified centres for the whole country with e.g. one providing FISH service, the other RNAseq. The information on methods provided is related to the results presented in Figure 1 to correlate with the data presented in Figure 1. This was the patient with a spectacular response and we wanted to cover the patient’s data more thoroughly. Optionally, it can be moved to figure legend or to a supplement, as further decided by reviewers and editor.

The remaining genetic data in the survey was obtained with standard methods as per treatment protocol and this is not elaborated on here. We now explain that fact.

Results:

I would generally not add comments in this section (e.g., “a surprisingly high proportion”) that need to be limited to the discussion

We modified the text accordingly

I would summarize genetic and cytogenetic alterations of included patients in the text

We added an appropriate summary.

The role of venetoclax combination therapies as bridge to CART cells therapy and HSCT should be reported in the text (eg proportion of patients bridged to HSCT and outcomes after transplant).

The proportion of patients that received HSCT /CAR-T or none of those after venetoclax is shown in Table 1. This survey covers a time when CAR-T used was not as widespread as currently and there is only one patient who received CAR-T so making comparisons and drawing conclusions would be very difficult.

It is also essential to mention in the results more details about the different therapies adopted in combination with venetoclax in the included patients, as this is a very important issue that is still unclear (e.g., azacytidine versus standard chemotherapy)

Table 1 lacks some essential information: prior therapies / HSCT; details regarding venetoclax cycles (number of cycles, combined therapies, responses after single cycles), HSCT/CART after venetoclax, relapse after venetoclax vs after CART/HSCT, causes of death

We believe a table summarising patient data and description of most significant patients is a reasonable compromise for the data to be presented. We believe extending much the table would make it impossible to follow. We propose not including detailed information on many different chemotherapy cycles used prior to venetoclax as this is not impact interpretation of venetoclax data and would make it difficult to follow. The descriptions of patients are already a bit lengthy. For some patients the final outcomes are not know at the time of data freeze. Also we believe subgrouping in 10 patients with different diseases by other features such as chemotherapy vs. demethylating agent leads to subgroups of 1-2 patients, this does not improve the report and give the reader more information, neither is plausible to statistical analysis.

The “single cases” description is interesting but maybe too hard to follow. I suggest summarizing data regarding different groups of diseases (e.g. ALL vs AML) with different responses.

Again we believe subgrouping of 10 patients with different diseases by other features leads to subgroups of 1-2 patients, this does not improve the report and give the reader more value. This will be possible with more widespread venetoclax use, especially if trail-based.

Lines 208-213: details regarding toxicity and complications related to venetoclax need to be reported

We argue reporting comorbidities would be misleading in our report in this heavily pretreated patient population, frequently with active neoplastic disease. No serious adverse event reports related to venetoclax were mentioned in patient medical data and thus it would be dubious to clearly ascribe toxicities to venetoclax. We mention this as limitation of our study. This would be more robust in the setting of clinical trials that our report may promote.

Lines 313-318 and Table 2: this part is interesting but maybe it does not fit with the rest of the paper. It can be maybe only mentioned when discussing responses in different genetic subtypes.

Regarding the DepMap-based table /paragraph we leave it do reviewers and editors decision whether this should be preserved, removed or perhaps moved to a supplement.

In the discussion I suggest summarizing the “review” part from lines 352 – 381 and discussing more in details the results of the study. It would be interesting to add a comparison of your results with the pediatric reports  published so far

We compared results of our study to major previous pediatric reports as requested. Unfortunately, our survey is neither a trial nor a prospective study, so generally we believe comparisons are very difficult, especially when it comes to any numerical approach.

Round 2

Reviewer 1 Report

The authors corrected the paper as suggested, therefore I do not have any other objections.

Author Response

We would like to express our sincere gratitude for the reviewer for the time and effort devoted to providing insights aimed at improving our manuscript.

Reviewer 2 Report

All the suggestions were made and answered.

The only concern is that "DepMap Cancer Dependency Map portal data access ofr BCL2 cell line sensitivity" does not add information to the manuscript. Only a search in the database was done and no substantial information is argued in the discussion section. Maybe this information can be suppressed and further deepened by the authors. Only mentioning which cell lines are from the pediatric is enough for discussion.

Please, at line 362, correct "ofr"

Author Response

We would like to express our sincere gratitude for the reviewers and editors for their time and effort devoted to providing insights aimed at improving our manuscript.

We corrected the typo and moved the DepMap data to a supplement.